# Self-supervised Co-training for Video Representation Learning

**Tengda Han, Weidi Xie, Andrew Zisserman**

VGG, Department of Engineering Science, University of Oxford

{htd, weidi, az}@robots.ox.ac.uk

http://www.robots.ox.ac.uk/~vgg/research/CoCLR/.

## Abstract

The objective of this paper is visual-only self-supervised video representation learning. We make the following contributions: (i) we investigate the benefit of adding semantic-class positives to instance-based Info Noise Contrastive Estimation (InfoNCE) training, showing that this form of supervised contrastive learning leads to a clear improvement in performance; (ii) we propose a novel *self-supervised co-training* scheme to improve the popular infoNCE loss, exploiting the complementary information from different views, RGB streams and optical flow, of the same data source by using one view to obtain positive class samples for the other; (iii) we thoroughly evaluate the quality of the learnt representation on two different downstream tasks: action recognition and video retrieval. In both cases, the proposed approach demonstrates state-of-the-art or comparable performance with other self-supervised approaches, whilst being significantly more efficient to train, *i.e.* requiring far less training data to achieve similar performance.

## 1 Introduction

The recent progress in self-supervised representation learning for images and videos has demonstrated the benefits of using a discriminative contrastive loss on data samples [12, 13, 27, 28, 45, 59], such as NCE [24, 34]. Given a data sample, the objective is to discriminate its transformed version against other samples in the dataset. The transformations can be artificial, such as those used in data augmentation [12], or natural, such as those arising in videos from temporal segments within the same clip. In essence, these pretext tasks focus on *instance discrimination*: each data sample is treated as a 'class', and the objective is to discriminate its own augmented version from a large number of other data samples or their augmented versions. Representations learned by instance discrimination in this manner have demonstrated extremely high performance on downstream tasks, often comparable to that achieved by supervised training [12, 27].

In this paper, we target self-supervised video representation learning, and ask the question: **is instance discrimination making the best use of data?** We show that the answer is *no*, in two respects:

First, we show that hard positives are being neglected in the self-supervised training, and that if these hard positives are included then the quality of learnt representation improves significantly. To investigate this, we conduct an *oracle* experiment where positive samples are incorporated into the instance-based training process based on the semantic class label. A clear performance gap is observed between the pure instance-based learning (termed *InfoNCE* [59]) and the oracle version (termed *UberNCE*). Note that the oracle is a form of supervised contrastive learning, as it encourages linear separability of the feature representation according to the class labels. In our experiments, training with UberNCE actually outperforms the supervised model trained with cross-entropy, a phenomenon that is also observed in a concurrent work [36] for image classification.

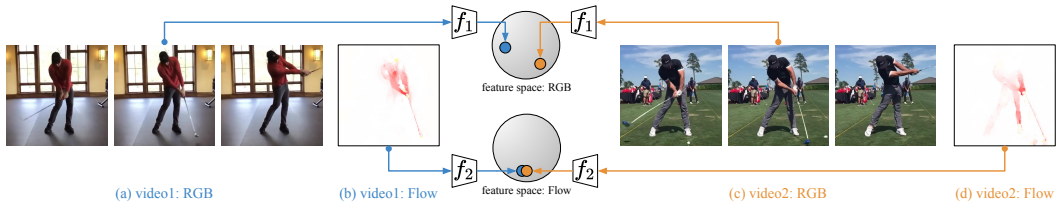

(a) video1: RGB      (b) video1: Flow      (c) video2: RGB      (d) video2: Flow

Figure 1: Two video clips of a golf-swing action and their corresponding optical flows. In this example, the flow patterns are very similar across different video instances despite significant variations in RGB space. This observation motivates the idea of co-training, which aims to gradually enhance the representation power of both networks, $f_1(\cdot)$ and $f_2(\cdot)$, by mining hard positives from one another.

Second, we propose a self-supervised co-training method, called *CoCLR*, standing for 'Co-training Contrastive Learning of visual Representation', with the goal of mining positive samples by using other complementary *views* of the data, *i.e.* replacing the role of the *oracle*. We pick RGB video frames and optical flow as the two views from hereon. As illustrated in Figure 1, positives obtained from flow can be used to 'bridge the gap' between the RGB video clips instances. In turn, positives obtained from RGB video clips can link optical flow clips of the same action. The outcome of training with the CoCLR algorithm is a representation that significantly surpasses the performance obtained by the instance-based training with InfoNCE, and approaches the performance of the *oracle* training with UberNCE.

To be clear, we are not proposing a new loss function or pretext task here, but instead we target the training regime by improving the sampling process in the contrastive learning of visual representation, *i.e.* constructing positive pairs beyond instances. There are two benefits: first, (hard) positive examples of the same class (*e.g.* the golf-swing action shown in Figure 1) are used in training; second, these positive samples are removed from the instance level negatives – where they would have been treated as false negatives for the action class. Our primary interest in this paper is to improve the representation of both the RGB and Flow networks, using the complementary information provided by the other view. For inference, we may choose to use only the RGB network or the Flow network, or both, depending on the applications and efficiency requirements.

To summarize, we investigate visual-only self-supervised video representation learning from RGB frames, or from unsupervised optical flow, or from both, and make the following contributions: (i) we show that an oracle with access to semantic class labels improves the performance of instance-based contrastive learning; (ii) we propose a novel self-supervised co-training scheme, CoCLR, to improve the training regime of the popular InfoNCE, exploiting the complementary information from different views of the same data source; and (iii) we thoroughly evaluate the quality of the learnt representation on two downstream tasks, video action recognition and retrieval, on UCF101 and HMDB51. In all cases, we demonstrate state-of-the-art or comparable performance over other self-supervised approaches, while being significantly more efficient, *i.e.* less data is required for self-supervised pre-training.

Our observations of using a second complementary view to bridge the RGB gap between positive instances from the same class is applied in this paper to optical flow. However, the idea is generally applicable for other complementary views: for videos, audio or text narrations can play a similar role to optical flow; whilst for still images, the multiple views can be formed by passing images through different filters. We return to this point in Section 5.

## 2   Related work

**Visual-only Self-supervised Learning.**  Self-supervised visual representation learning has recently witnessed rapid progress in image classification. Early work in this area defined proxy tasks explicitly, for example, colorization, inpainting, and jigsaw solving [15, 16, 48, 68]. More recent approaches jointly optimize clustering and representation learning [5, 9, 10] or learn visual representation by discriminating instances from each other through contrastive learning [12, 27, 28, 29, 32, 45, 57, 59, 70]. Videos offer additional opportunities for learning representations, beyond those of images, by exploiting spatio-temporal information, for example, by ordering frames or clips [21, 42, 46, 64, 66],

motion [1, 14, 31], co-occurrence [30], jigsaw [37], rotation [33], speed prediction [6, 17, 62], future prediction [25, 26, 60], or by temporal coherence [40, 41, 61, 63].

**Multi-modal Self-supervised Learning.** This research area focuses on leveraging the interplay of different modalities, for instance, contrastive loss is used to learn the correspondence between frames and audio [2, 3, 4, 38, 49, 50], or video and narrations [44]; or, alternatively, an iterative clustering and re-labelling approach for video and audio has been used in [2].

**Co-training Paired Networks.** Co-training [7] refers to a semi-supervised learning technique that assumes each example to be described by multiple views that provide different and complementary information about the instance. Co-training first learns a separate classifier for each view using any labelled examples, and the most confident predictions of each classifier on the unlabelled data are then used to iteratively construct additional labelled training data. Though note in our case that we have *no* labelled samples. More generally, the idea of having two networks interact and co-train also appears in other areas of machine learning, *e.g.* Generative Adversarial Networks (GANs) [22], and Actor-Critic Reinforcement Learning [56].

**Video Action Recognition.** This research area has gone through a rapid development in recent years, from the two-stream networks [14, 20, 52, 69] to the more recent single stream RGB networks [18, 19, 58, 65], and the action classification performance has steadily improved. In particular, the use of distillation [14, 54], where the flow-stream network is used to teach the RGB-stream network, at a high level, is related to the goal in this work.

## 3  InfoNCE, UberNCE and CoCLR

We first review instance discrimination based self-supervised learning with InfoNCE, as used by [12], and introduce an *oracle* extension where positive samples are incorporated into the instance-based training process based on the semantic class label. Then in Section 3.2, we introduce the key idea of mining informative positive pairs using multiview co-training, and describe our algorithm for employing it, which enables InfoNCE to extend beyond instance discrimination.

### 3.1  Learning with InfoNCE and UberNCE

**InfoNCE.** Given a dataset with $N$ raw video clips, *e.g.* $\mathcal{D} = \{x_1, x_2, \ldots, x_N\}$, the objective for self-supervised video representation learning is to obtain a function $f(\cdot)$ that can be effectively used to encode the video clips for various downsteam tasks, *e.g.* action recognition, retrieval, *etc.*

Assume there is an augmentation function $\psi(\cdot; a)$, where $a$ is sampled from a set of pre-defined data augmentation transformations $A$, that is applied to $\mathcal{D}$. For a particular sample $x_i$, the positive set $\mathcal{P}_i$ and the negative set $\mathcal{N}_i$ are defined as: $\mathcal{P}_i = \{\psi(x_i; a) | a \sim A\}$, and $\mathcal{N}_i = \{\psi(x_n; a) | \forall n \neq i, a \sim A\}$. Given $z_i = f(\psi(x_i; \cdot))$, then the InfoNCE loss is:

$$\mathcal{L}_{\text{InfoNCE}} = -\mathbb{E}\left[\log \frac{\exp(z_i \cdot z_p / \tau)}{\exp(z_i \cdot z_p / \tau) + \sum_{n \in \mathcal{N}_i} \exp(z_i \cdot z_n / \tau)}\right] \tag{1}$$

where $z_i \cdot z_p$ refers to the dot product between two vectors. In essence, the objective for optimization can be seen as instance discrimination, *i.e.* emitting higher similarity scores between the augmented views of the *same* instance than with augmented views from *other* instances.

**UberNCE.** Assume we have a dataset with annotations, $\mathcal{D} = \{(x_1, y_1), (x_2, y_2), \ldots, (x_N, y_N)\}$, where $y_i$ is the class label for clip $x_i$, and an *oracle* that has access to these annotations. We search for a function $f(\cdot)$, by optimizing an identical InfoNCE to Eq. 1, *except* that for each sample $x_i$, the positive set $\mathcal{P}_i$ and the negative set $\mathcal{N}_i$ can now include samples with same semantic labels, in addition to the augmentations, *i.e.* $\mathcal{P}_i = \{\psi(x_i; a), x_p | y_p = y_i \text{ and } p \neq i, \forall p \in [1, N], a \sim A\}$, $\mathcal{N}_i = \{\psi(x_n; a), x_n | y_n \neq y_i, \forall n \in [1, N], a \sim A\}$.

As an example, given an input video clip of a 'running' action, the positive set contains its own augmented version and all other 'running' video clips in the dataset, and the negative set consists all video clips from other action classes.

As will be demonstrated in Section 4.4, we evaluate the representation on a linear probe protocol, and observe a significant performance gap between training on InfoNCE and UberNCE, confirming that

**instance discrimination is not making the best use of data.** Clearly, the choice of sampling more informative *positives i.e.* treating semantically related clips as positive pairs (and thereby naturally eliminating *false negatives*), plays a vital role in such representation learning, as this is the *only* difference between InfoNCE and UberNCE.

## 3.2 Self-supervised CoCLR

As an extension of the previous notation, given a video clip $x_i$, we now consider two different views, $\mathbf{x}_i = \{x_{1i}, x_{2i}\}$, where in this paper, $x_{1i}$ and $x_{2i}$ refer to RGB frames and their *unsupervised* optical flows respectively. The objective of self-supervised video representation learning is to learn the functions $f_1(\cdot)$ and $f_2(\cdot)$, where $z_{1i} = f_1(x_{1i})$ and $z_{2i} = f_2(x_{2i})$ refer to the representations of the RGB stream and optical flow, that can be effectively used for performing various downstream tasks.

The key idea, and how the method differs from InfoNCE and UberNCE, is in the construction of the positive set ($\mathcal{P}_i$) and negative set ($\mathcal{N}_i$) for the sample $x_i$. Intuitively, positives that are very hard to 'discover' in the RGB stream can often be 'easily' determined in the optical flow stream. For instance, under static camera settings, flow patterns from a particular action, such as golf swing, can be very similar across instances despite significant background variations that dominate the RGB representation (as shown in Figure 1). Such similarities can be discovered even with a partially trained optical flow network. This observation enables two models, one for RGB and the other for flow, to be co-trained, starting from a bootstrap stage and gradually enhancing the representation power of both as the training proceeds.

In detail, we co-train the models by mining positive pairs from the other view of data. The RGB representation $f_1(\cdot)$ is updated with a Multi-Instance InfoNCE [44] loss (that covers our case of one or more actual positives within the positive set $\mathcal{P}_{1i}$ defined below):

$$\mathcal{L}_1 = -\mathbb{E}\left[\log \frac{\sum_{p\in\mathcal{P}_{1i}} \exp(z_{1i} \cdot z_p/\tau)}{\sum_{p\in\mathcal{P}_{1i}} \exp(z_{1i} \cdot z_p/\tau) + \sum_{n\in\mathcal{N}_{1i}} \exp(z_{1i} \cdot z_n/\tau)}\right] \quad (2)$$

where the numerator is defined as a sum of 'similarity' between sample $x_{1i}$ (in the RGB view) and a positive set, constructed by the video clips that are most similar to $x_{2i}$ (most similar video clips in the optical flow view):

$$\mathcal{P}_{1i} = \{\psi(x_{1i}; a), x_{1k} | k \in \text{topK}(z_{2i} \cdot z_{2j}), \forall j \in [1, N], a \sim A\} \quad (3)$$

$z_{2i} \cdot z_{2j}$ refers to the similarity between $i$-th and $j$-th video in the optical flow view, and the topK$(\cdot)$ operator selects the top $K$ items over all available $N$ samples and returns their indexes. The $K$ is a hyper parameter representing the strictness of positive mining. The negative set $\mathcal{N}_{1i}$ for sample $x_i$ is the complement of the positive set, $\mathcal{N}_{1i} = \overline{\mathcal{P}_{1i}}$. In other words, the positive set consists of the top $K$ nearest neighbours in the optical flow feature space plus the video clip's own augmentations, and the negative set contains all other video clips, and their augmentations.

Similarly, to update the optical flow representation, $f_2(\cdot)$, we can optimize:

$$\mathcal{L}_2 = -\mathbb{E}\left[\log \frac{\sum_{p\in\mathcal{P}_{2i}} \exp(z_{2i} \cdot z_p/\tau)}{\sum_{p\in\mathcal{P}_{2i}} \exp(z_{2i} \cdot z_p/\tau) + \sum_{n\in\mathcal{N}_{2i}} \exp(z_{2i} \cdot z_n/\tau)}\right] \quad (4)$$

It is an identical objective function to (2) except that the positive set is now constructed from similarity ranking in the RGB view:

$$\mathcal{P}_{2i} = \{\psi(_{2i}; a), x_{2k} | k \in \text{topK}(z_{1i} \cdot z_{1j}), \forall j \in [1, N], a \sim A\} \quad (5)$$

**The CoCLR algorithm**   proceeds in two stages: initialization and alternation.

*Initialization.* To start with, the two models with different views are trained independently with InfoNCE, *i.e.* the RGB and Flow networks are trained by optimizing $\mathcal{L}_{\text{InfoNCE}}$.

*Alternation.* Once trained with $\mathcal{L}_{\text{InfoNCE}}$, both the RGB and Flow networks have gained far stronger representations than randomly initialized networks. The co-training process then proceeds as described in Eq. 2 and Eq. 4, *e.g.* to optimize $\mathcal{L}_1$, we mine hard positive pairs with a Flow network; to optimize $\mathcal{L}_2$, the hard positive pairs are mined with a RGB network. These two optimizations are

alternated: each time first mining hard positives from the other network, and then minimizing the loss for the network independently. As the joint optimization proceeds, and the representations become stronger, different (and harder) positives are retrieved.

The key hyper-parameters that define the alternation process are: the value of $K$ used to retrieve the $K$ semantically related video clip, and the number of iterations (or epochs) to minimize each loss function, *i.e.* the granularity of the alternation. These choices are explored in the ablations of Section 4.4, where it will be seen that a choice of $K = 5$ is optimal and more cycle alternations are beneficial; where each cycle refers to a complete optimization of $\mathcal{L}_1$ and $\mathcal{L}_2$; meaning, the alternation only happens after the RGB or Flow network has converged.

**Discussion.** *First*, when compared with our previous work that used InfoNCE for video self-supervision, DPC and MemDPC [25, 26], the proposed CoCLR incorporates learning from potentially harder positives, *e.g.* instances from the same class, rather than from only different augmentations of the same instance; *Second*, CoCLR differs from the *oracle* proposals of UberNCE since both the CoCLR positive and negative sets may still contain 'label' noise, *i.e.* class-wise false positives and false negatives. However, in practice, the Multi-Instance InfoNCE used in CoCLR is fairly robust to noise. *Third*, CoCLR is fundamentally different to two concurrent approaches, CMC [57] and CVRL [51], that use only instance-level training, *i.e.* positive pairs are constructed from the same data sample. Specifically, CMC extends positives to include different views, RGB and flow, of the same video clip, but does not introduce positives between clips; CVRL uses InfoNCE contrastive learning with video clips as the instances. We present experimental results for both InfoNCE and CMC in Table 1.

## 4 Experiments

In this section, we first describe the datasets (Section 4.1) and implementation details (Section 4.2) for CoCLR training. In Section 4.3, we describe the downstream tasks for evaluating the representation obtained from self-supervised learning. All proof-of-concept and ablation studies are conducted on UCF101 (Section 4.4), with larger scale training on Kinetics-400 (Section 4.5) to compare with other state-of-the-art approaches.

### 4.1 Datasets

We use two video action recognition datasets for self-supervised CoCLR training: **UCF101** [53], containing 13k videos spanning 101 human actions (we only use the videos from the training set); and Kinetics-400 (**K400**) [35] with 240k video clips only from its training set. For downstream evaluation tasks, we benchmark on the UCF101 split1, K400 validation set, as well as on the split1 of **HMDB51** [39], which contains 7k videos spanning 51 human actions.

### 4.2 Implementation Details for CoCLR

We choose the S3D [65] architecture as the feature extractor for all experiments. During CoCLR training, we attach a non-linear projection head, and remove it for downstream task evaluations, as done in SimCLR [12]. We use a 32-frame RGB (or flow) clip as input, at 30 fps, this roughly covers 1 second. The video clip has a spatial resolution of $128 \times 128$ pixels. For data augmentation, we apply random crops, horizontal flips, Gaussian blur and color jittering, all are clip-wise consistent. We also apply random temporal cropping to utilize the natural variation of the temporal dimension, *i.e.* the input video clips are cropped at random time stamps from the source video. The optical flow is computed with the *un-supervised* TV-L1 algorithm [67], and the same pre-processing procedure is used as in [11]. Specifically, two-channel motion fields are stacked with a third zero-valued channel, large motions exceeding 20 pixels are truncated, and the values are finally projected from $[-20, 20]$ to $[0, 255]$ then compressed as jpeg.

At the *initialization* stage, we train both RGB and Flow networks with InfoNCE for 300 epochs, where an epoch means to have sampled one clip from each video in the training set, *i.e.* the total number of seen instances is equivalent to the number of videos in the training set. We adopt a momentum-updated history queue to cache a large number of features as in MoCo [13, 27]. At the *alternation* stage, on UCF101 the model is trained for two cycles, where each cycle includes 200 epochs, *i.e.* RGB and Flow networks are each trained for 100 epochs with hard positive mining from

the other; on K400 the model is only trained for one cycle for 100 epochs, that is 50 epochs each for RGB and Flow networks, however, we expect more training cycles to be beneficial. For optimization, we use Adam with $10^{-3}$ learning rate and $10^{-5}$ weight decay. The learning rate is decayed down by $1/10$ twice when the validation loss plateaus. Each experiment is trained on 4 GPUs, with a batch size of 32 samples per GPU.

### 4.3 Downstream tasks for representation evaluation

**Action classification.** In this protocol, we evaluate on two settings: (1) **linear probe**: the entire feature encoder is frozen, and only a single linear layer is trained with cross-entropy loss, (2) **finetune**: the entire feature encoder and a linear layer are finetuned end-to-end with cross-entropy loss, *i.e.* the representation from CoCLR training provides an initialization for the network.

At the training stage, we apply the same data augmentation as in the pre-training stage mentioned in Section 4.2, except for the Gaussian blur. At the inference stage, we follow the same procedure as our previous work [25, 26]: for each video we spatially apply ten-crops (center crop plus four corners, with horizontal flipping) and temporally take clips with moving windows (half temporal overlap), and then average the predicted probabilities.

**Action retrieval.** In this protocol, the extracted feature is directly used for nearest-neighbour (NN) retrieval and no further training is allowed. We follow the common practice [43, 66], and use testing set video clips to query the $k$-NNs from the training set. We report Recall at $k$ (R@$k$), meaning, if the top $k$ nearest neighbours contains one video of the same class, a correct retrieval is counted.

### 4.4 Model comparisons on UCF101

This section demonstrates the evolution from InfoNCE to UberNCE and to CoCLR, and we monitor the top1 accuracy of action classification and retrieval performance. In this section, the *same* dataset, UCF101 split # 1 is used for self-supervised training and downstream evaluations, and we mainly focus on the linear probe & retrieval as the primary measures of representation quality, since their evaluation is fast. For all self-supervised pretraining, we keep the settings identical, *e.g.* training epochs, and only vary the process for mining positive pairs.

| Pretrain Stage | | | Classification Top1 | | Retrieval |
|---|---|---|---|---|---|
| Method | Input | Labels | Linear probe | Finetune | R@1 |
| InfoNCE | RGB | ✗ | 46.8 | 78.4 | 33.1 |
| InfoNCE | Flow | ✗ | 66.8 | 83.1 | 45.2 |
| UberNCE | RGB | ✓ | 78.0 | 80.0 | 71.6 |
| Cross-Ent. | RGB | ✓ | - | 77.0* | 73.5 |
| CMC§ [57] | RGB | ✓ | 55.0 | - | - |
| CoCLR$_{K=5}$ | RGB | ✗ | 70.2 | 81.4 | 51.8 |
| CoCLR$_{K=5}$ | Flow | ✗ | 68.7 | 83.5 | 48.4 |
| CoCLR$_{K=5}$† | R+F | ✗ | **72.1** | **87.3** | **55.6** |
| CoCLR$_{K=1}$ | RGB | ✗ | 60.5 | 79.5 | 48.5 |
| CoCLR$_{K=50}$ | RGB | ✗ | 68.3 | 81.0 | 49.8 |
| CoCLR$_{K=5, sim}$ | RGB | ✗ | 65.2 | 80.8 | 48.0 |

Table 1: Representations from InfoNCE, UberNCE and Co-CLR are evaluated on downstream action classification and retrieval. Left refers to the setting for pre-training. CMC§ is our implementation for a fair comparison to CoCLR, *i.e.* S3D architecture, trained with 500 epochs. † refers to the results from two-stream networks (RGB + Flow). *Cross-Ent. is end-to-end training with Softmax Cross-Entropy.

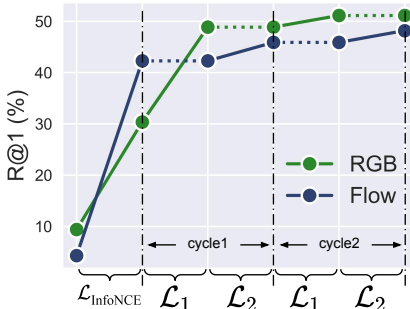

Figure 2: Training progress of CoCLR on UCF101 dataset for RGB and optical flow input, *i.e.* R@1 of training set video retrieval for both RGB and Flow, the dotted line means that the representation is fixed at certain training stage.

**Linear probe & retrieval.** The discussion here will focus on the RGB network, as this network is easy to use (no flow computation required) and offers fast inference speed, but training was done with both RGB and Flow for CoCLR and CMC. As shown in Table 1, three phenomena can be observed: *First*, UberNCE, the *supervised* contrastive method outperforms the InfoNCE baseline with a significant gap on the linear probe (78.0 vs 46.8) and retrieval (71.6 vs 33.1), which reveals the suboptimality of the *instance-based* self-supervised learning. *Second*, the co-training scheme (CoCLR) shows its effectiveness by substantially improving over the InfoNCE and CMC baselines from 46.8 and 55.0 to 70.2, approaching the results of UberNCE (78.0). *Third*, combining

the logits from both the RGB and Flow networks (denoted as CoCLR†) brings further benefits. We conjecture that a more modern RGB network, such as SlowFast [19], that is able to naturally capture more of the motion information, would close the gap even further.

**End-to-end finetune.** In this protocol, all models (RGB networks) are performing similarly well, and the gaps between different training schemes are marginal (77.0 – 81.4). This is expected, as the same dataset, data view and architecture have been used for self-supervised learning, finetuning or training from scratch. In this paper, we are more interested in the scenario, where pretraining is conducted on a large-scale dataset, *e.g.* Kinetics, and feature transferability is therefore evaluated on linear probing and finetuning on another small dataset, as demonstrated in Section 4.5.

For a better understanding of the effectiveness of co-training on mining hard positive samples, in Figure 2, we monitor the *alternation* process by measuring R@1. Note that, the label information is only used to plot this curve, but *not* used during self-supervised training. The x-axis shows the training stage, initialized from the InfoNCE representation, followed by alternatively training $\mathcal{L}_1$ and $\mathcal{L}_2$ for two cycles, as explained in Section 3.2. The dotted line indicates that a certain network is fixed, and the solid line indicates that the representation is being optimized. As training progresses, the representation quality of both the RGB and Flow models improve with more co-training cycles, shown by the increasing R@1 performance, which indicates that the video clips with same class have indeed been pulled together in the embedding space.

**Ablations.** We also experimented with other choices for the CoCLR hyper-parameters, and report the results at the bottom Table 1. In terms of number of the samples mined in Eq. 3 and Eq. 5, $K = 5$ is the optimal setting, *i.e.* the the Top5 most similar samples are used to train the target representation. Other values, $K = 1$ and $K = 50$ are slightly worse. In terms of *alternation* granularity, we compare with the extreme case that the two representations are optimized simultaneously ($\text{CoCLR}_{K=5;\,\text{sim}}$), again, this performs slightly worse than training one network with the other fixed to 'act as' an oracle. We conjecture that the inferior performance of simultaneous optimization is because the weights of the network are updated too fast, similar phenomena have also been observed in other works [27, 56], we leave further investigation of this to future work.

### 4.5 Comparison with the state-of-the-art

In this section, we compare CoCLR with previous self-supervised approaches on action classification. Specifically, we provide results of CoCLR under two settings, namely, trained on UCF101 with $K = 5$ for 2 cycles; and on K400 with $K = 5$ for 1 cycle only. Note that there has been a rich literature on video self-supervised learning, and in Table 2 we only list some of the recent approaches evaluated on the same benchmark, and try to compare with them as fairly as we can, in terms of architecture, training data, resolution (although there remain variations).

In the following we compare with the methods that are trained with: (i) visual information only on the same training set; (ii) visual information only on larger training sets; (iii) multimodal information.

**Visual-only information with same training set (finetune).** When comparing the models that are only trained (both self-supervised and downstream finetune) on **UCF101**, *e.g.* OPN and VCOP, the proposed CoCLR (RGB network) obtains Top1 accuracy of 81.4 on UCF101 and 52.1 on HMDB, significantly outperforming all previous approaches. Moving onto **K400**, recent approaches include 3D-RotNet, ST-Puzzle, DPC, MemDPC, XDC, GDT, and SpeedNet. Again, CoCLR (RGB network) surpasses the other self-supervised methods, achieving 87.9 on UCF101 and 54.6 on HMDB, and the two-stream CoCLR† brings further benefits (90.6 on UCF101 and 62.9 on HMDB). We note that CoCLR is slightly underperforming CVRL [51], which we conjecture is due to the fact that CVRL has been trained with a deeper architecture (23 vs. 49 layers) with more parameters (7.9M vs. 33.1M), larger resolution (128 vs. 224), stronger color jittering, and far more epochs (400 vs. 800 epochs). This also indicates that potentially there remains room for further improving CoCLR, starting from a better initialization trained with InfoNCE.

**Visual-only information with larger training set (finetune).** Although only visual information is used, some approaches exploit a larger training set, *e.g.* CBT and DynamoNet. CoCLR† still outperforms all these approaches, showing its remarkable training efficiency, in the sense that it can learn better representation with far less data.

**Multi-modal information (finetune).** These are the methods that exploit the correspondence of visual information with text or audio. The methods usually train on much larger-scale datasets,

for instance, AVTS trained on AudioSet (8x larger than K400), and XDC trained on IG65M (273x larger than K400), for audio-visual correspondence; MIL-NCE is trained on narrated instructional videos (195x larger than K400) for visual-text correspondence; and ELO [50] is trained with 7 different losses on 2 million videos (104x larger than K400). Despite these considerably larger datasets, and information from other modalities, our best visual-only CoCLR† (two-stream network) still compares favorably with them. Note that, our CoCLR approach is also not limited to visual-only self-supervised learning, and is perfectly applicable for mining hard positives from audio or text.

| Method | Date | Dataset (duration) | Res. | Arch. | Depth | Modality | Frozen | UCF | HMDB |
|---|---|---|---|---|---|---|---|---|---|
| CBT [55] | 2019 | K600+ (273d) | 112 | S3D | 23 | V | ✓ | 54.0 | 29.5 |
| MemDPC [26] | 2020 | K400 (28d) | 224 | R-2D3D | 33 | V | ✓ | 54.1 | 30.5 |
| MIL-NCE [44] | 2020 | HTM (15y) | 224 | S3D | 23 | V+T | ✓ | 82.7 | 53.1 |
| MIL-NCE [44] | 2020 | HTM (15y) | 224 | I3D | 22 | V+T | ✓ | 83.4 | 54.8 |
| XDC [2] | 2019 | IG65M (21y) | 224 | R(2+1)D | 26 | V+A | ✓ | 85.3 | 56.0 |
| ELO [50] | 2020 | Youtube8M- (8y) | 224 | R(2+1)D | 65 | V+A | ✓ | – | 64.5 |
| **CoCLR**-RGB | | UCF (1d) | 128 | S3D | 23 | V | ✓ | 70.2 | 39.1 |
| **CoCLR**-2Stream† | | UCF (1d) | 128 | S3D | 23 | V | ✓ | 72.1 | 40.2 |
| **CoCLR**-RGB | | K400 (28d) | 128 | S3D | 23 | V | ✓ | 74.5 | 46.1 |
| **CoCLR**-2Stream† | | K400 (28d) | 128 | S3D | 23 | V | ✓ | 77.8 | 52.4 |
| OPN [42] | 2017 | UCF (1d) | 227 | VGG | 14 | V | ✗ | 59.6 | 23.8 |
| 3D-RotNet [33] | 2018 | K400 (28d) | 112 | R3D | 17 | V | ✗ | 62.9 | 33.7 |
| ST-Puzzle [37] | 2019 | K400 (28d) | 224 | R3D | 17 | V | ✗ | 63.9 | 33.7 |
| VCOP [66] | 2019 | UCF (1d) | 112 | R(2+1)D | 26 | V | ✗ | 72.4 | 30.9 |
| DPC [25] | 2019 | K400 (28d) | 128 | R-2D3D | 33 | V | ✗ | 75.7 | 35.7 |
| CBT [55] | 2019 | K600+ (273d) | 112 | S3D | 23 | V | ✗ | 79.5 | 44.6 |
| DynamoNet [14] | 2019 | Youtube8M-1 (58d) | 112 | STCNet | 133 | V | ✗ | 88.1 | 59.9 |
| SpeedNet [6] | 2020 | K400 (28d) | 224 | S3D-G | 23 | V | ✗ | 81.1 | 48.8 |
| MemDPC [26] | 2020 | K400 (28d) | 224 | R-2D3D | 33 | V | ✗ | 86.1 | 54.5 |
| CVRL [51] | 2020 | K400 (28d) | 224 | R3D | 49 | V | ✗ | 92.1 | 65.4 |
| AVTS [38] | 2018 | K400 (28d) | 224 | I3D | 22 | V+A | ✗ | 83.7 | 53.0 |
| AVTS [38] | 2018 | AudioSet (240d) | 224 | MC3 | 17 | V+A | ✗ | 89.0 | 61.6 |
| XDC [2] | 2019 | K400 (28d) | 224 | R(2+1)D | 26 | V+A | ✗ | 84.2 | 47.1 |
| XDC [2] | 2019 | IG65M (21y) | 224 | R(2+1)D | 26 | V+A | ✗ | 94.2 | 67.4 |
| GDT [49] | 2020 | K400 (28d) | 112 | R(2+1)D | 26 | V+A | ✗ | 89.3 | 60.0 |
| GDT [49] | 2020 | G65M (21y) | 112 | R(2+1)D | 26 | V+A | ✗ | 95.2 | 72.8 |
| MIL-NCE [44] | 2020 | HTM (15y) | 224 | S3D | 23 | V+T | ✗ | 91.3 | 61.0 |
| ELO [50] | 2020 | Youtube8M-2 (13y) | 224 | R(2+1)D | 65 | V+A | ✗ | 93.8 | 67.4 |
| **CoCLR**-RGB | | UCF (1d) | 128 | S3D | 23 | V | ✗ | 81.4 | 52.1 |
| **CoCLR**-2Stream† | | UCF (1d) | 128 | S3D | 23 | V | ✗ | 87.3 | 58.7 |
| **CoCLR**-RGB | | K400 (28d) | 128 | S3D | 23 | V | ✗ | 87.9 | 54.6 |
| **CoCLR**-2Stream† | | K400 (28d) | 128 | S3D | 23 | V | ✗ | 90.6 | 62.9 |
| Supervised [65] | | K400 (28d) | 224 | S3D | 23 | V | ✗ | 96.8 | 75.9 |

Table 2: Comparison with state-of-the-art approaches. In the left columns, we show the pre-training setting, *e.g.* dataset, resolution, architectures with encoder depth, modality. In the right columns, the top-1 accuracy is reported on the downstream action classification task for different datasets, *e.g.* UCF, HMDB, K400. The dataset parenthesis shows the total video duration in time (**d** for day, **y** for year). 'Frozen ✗' means the network is end-to-end finetuned from the pretrained representation, shown in the top half of the table; 'Frozen ✓' means the pretrained representation is fixed and classified with a linear layer, shown in the bottom half. For input, 'V' refers to visual only (colored with blue), 'A' is audio, 'T' is text narration. CoCLR models with † refer to the two-stream networks, where the predictions from RGB and Flow networks are averaged.

**Linear probe.** As shown in the upper part of Table 2, CoCLR outperforms MemDPC and CBT significantly, with the same or only a tiny proportion of data for self-supervised training, and compares favorably with MIL-NCE, XDC and ELO that are trained on orders of magnitude more training data.

**Video retrieval.** In addition to the action classification benchmarks, we also evaluate CoCLR on video retrieval, as explained in Section 4.3. The goal is to test if the query clip instance and its nearest neighbours belong to same semantic category. As shown in Table 3, in both benchmark datasets, the InfoNCE baseline models exceed all previous approaches by a significant margin. Our CoCLR models further exceed InfoNCE models by a large margin.

**Qualitative results for video retrieval.** Figure 3 visualizes a query video clip and its Top3 Nearest Neighbors from the UCF101 training set using the CoCLR embedding. As can be seen, the representation learnt by CoCLR has the ability to retrieve videos with the same semantic categories.

| Method | Date | Dataset | UCF | | | | HMDB | | | |
|---|---|---|---|---|---|---|---|---|---|---|
| | | | R@1 | R@5 | R@10 | R@20 | R@1 | R@5 | R@10 | R@20 |
| Jigsaw [47] | 2016 | UCF | 19.7 | 28.5 | 33.5 | 40.0 | - | - | - | - |
| OPN [42] | 2017 | UCF | 19.9 | 28.7 | 34.0 | 40.6 | - | - | - | - |
| Buchler [8] | 2018 | UCF | 25.7 | 36.2 | 42.2 | 49.2 | - | - | - | - |
| VCOP [66] | 2019 | UCF | 14.1 | 30.3 | 40.4 | 51.1 | 7.6 | 22.9 | 34.4 | 48.8 |
| VCP [43] | 2020 | UCF | 18.6 | 33.6 | 42.5 | 53.5 | 7.6 | 24.4 | 36.3 | 53.6 |
| MemDPC [26] | 2020 | UCF | 20.2 | 40.4 | 52.4 | 64.7 | 7.7 | 25.7 | 40.6 | 57.7 |
| SpeedNet [6] | 2020 | K400 | 13.0 | 28.1 | 37.5 | 49.5 | - | - | - | - |
| InfoNCE-RGB | | UCF | 36.0 | 52.0 | 61.8 | 71.0 | 15.2 | 34.7 | 48.9 | 63.2 |
| InfoNCE-Flow | | UCF | 45.5 | 67.5 | 75.4 | **82.7** | 21.4 | 46.3 | **59.6** | **72.1** |
| **CoCLR**-RGB | | UCF | 53.3 | 69.4 | 76.6 | 82.0 | 23.2 | 43.2 | 53.5 | 65.5 |
| **CoCLR**-Flow | | UCF | 51.9 | 68.5 | 75.0 | 80.8 | 23.9 | **47.3** | 58.3 | 69.3 |
| **CoCLR**-2Stream† | | UCF | **55.9** | **70.8** | **76.9** | 82.5 | **26.1** | 45.8 | 57.9 | 69.7 |

Table 3: Comparison with others on Nearest-Neighbour video retrieval on UCF101 and HMDB51. Testing set clips are used to retrieve training set videos and R@$k$ is reported, where $k \in [1, 5, 10, 20]$. Note that all the models reported were only pretrained on UCF101 with self-supervised learning except SpeedNet. † For two-stream network, the feature similarity scores from RGB and Flow networks are averaged.

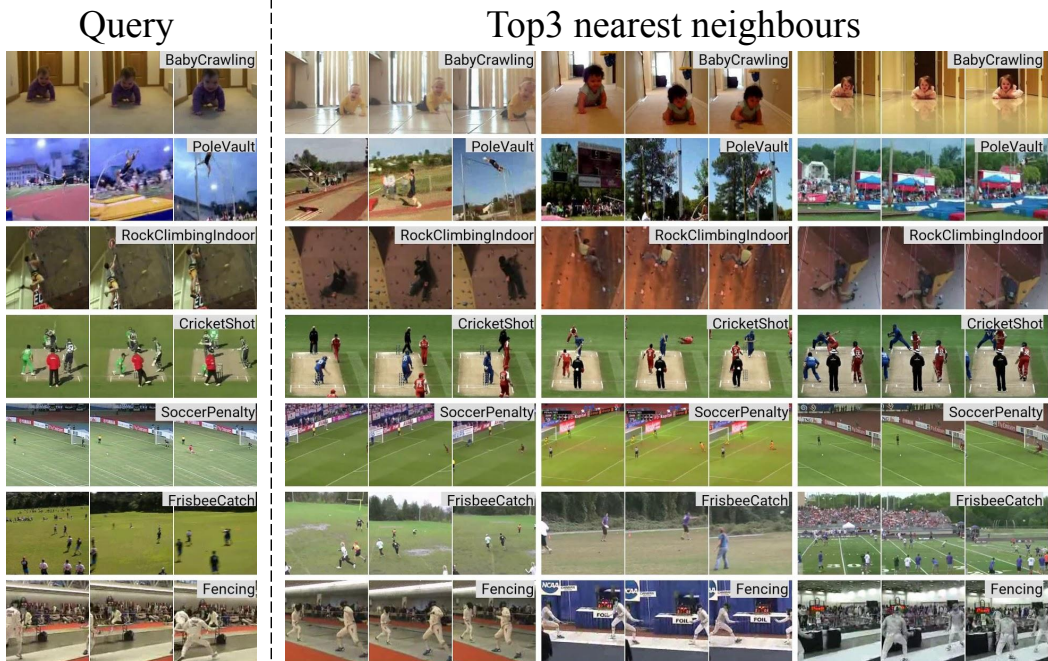

Figure 3: Nearest neighbour retrieval results with CoCLR representations. The left side is the query video from the UCF101 testing set, and the right side are the top 3 nearest neighbours from the UCF101 training set. CoCLR is trained only on UCF101. The action label for each video is shown in the upper right corner.

## 5 Conclusion

We have shown that a complementary view of video can be used to bridge the gap between RGB video clip instances of the same class, and that using this to generate positive training sets substantially improves the performance over InfoNCE instance training for video representations. Though we have not shown it in this paper, we conjecture that *explicit* mining from audio can provide a similar role to optical flow. For example, the sound of a guitar can link together video clips with very different visual appearances, even if the audio network is relatively untrained. This observation in part explains the success of audio-visual self-supervised learning, *e.g.* [2, 3, 4, 38, 49] where such links occur *implicitly*. Similarly and more obviously, text provides the bridge between instances in visual-text learning, *e.g.* from videos with narrations that describe their visual content [44]. We expect that the success of explicit positive mining in CoCLR will lead to applications to other data, *e.g.* images, other modalities and tasks where other views can be extracted to provide complementary information, and also to other learning methods, such as BYOL [23].

## 6 Broader Impact

Deep learning systems are data-hungry and are often criticized for their huge financial and environmental cost. Training a deep neural network end-to-end is especially expensive due to the large computational requirements. Our research on video representation learning has shown its effectiveness on various downstream tasks. As a positive effect of this, future research can benefit from our work by building systems with the pretrained representation to save the cost of re-training. However, on the negative side, research on self-supervised representation learning has consumed many computational resources and we hope more efforts are put on reducing the training cost in this research area. To facilitate future research, we release our code and pretrained representations.

## Acknowledgement

We thank Triantafyllos Afouras, Andrew Brown, Christian Rupprecht and Chuhan Zhang for proof-reading and helpful discussions. Funding for this research is provided by a Google-DeepMind Graduate Scholarship, and by the EPSRC Programme Grant Seebibyte EP/M013774/1.

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
