[Supplementary Material]

# Supplementary Material for
## Self-supervised Co-Training for Video Representation Learning

**Tengda Han, Weidi Xie, Andrew Zisserman**
VGG, Department of Engineering Science
University of Oxford
{htd, weidi, az}@robots.ox.ac.uk

## 1 More Implementation Details

### 1.1 Encoder Architecture

We use the S3D architecture for all experiments. At the pretraining stage (including InfoNCE and CoCLR), S3D is followed by a non-linear projection head. Specifically, the project head consists of two fully-connected (FC) layers. The projection head is removed when evaluating downstream tasks. The detailed dimensions are shown in Table 1.

| Stage | Detail | Output size: T×HW×C |
|---|---|---|
| S3D | followed by average pooling | $1 \times 1^2 \times 1024$ |
| Projection head | FC-1024→ReLU→FC-128 | $1 \times 1^2 \times 128$ |

Table 1: Feature encoder architecture at the pretraining stage. 'FC-1024' and 'FC-128' denote the output dimension of each fully-connected layer respectively.

### 1.2 Classifier Architecture

When evaluating the pretrained representation for action classification, we replace the non-linear projection head with a single linear layer for the classification tasks. The detailed dimensions are shown in Table 2.

| Stage | Detail | Output size: T×HW×C |
|---|---|---|
| S3D | followed by average pooling | $1 \times 1^2 \times 1024$ |
| Linear layer | one layer: FC-num_class | $1 \times 1^2 \times$ num_class |

Table 2: Classifier architecture for evaluating the representation on action classification tasks. 'FC-num_class' denotes the output dimension of fully-connected layer is the number of action classes.

### 1.3 Momentum-updated History Queue

To cache a large number of features, we adopt a momentum-updated history queue as in MoCo [1]. The history queue is used in all pretraining experiments (including both InfoNCE and CoCLR). For the pretraining on UCF101, we use softmax temperature $\tau = 0.07$, momentum $m = 0.999$ and queue size 2048; for the pretraining on K400, we use softmax temperature $\tau = 0.07$, momentum $m = 0.999$ and queue size 16384.

# 2 Example Code for CoCLR

In this section, we give an example implementation of CoCLR in PyTorch-like style for training $\mathcal{L}_1$ in Eq.2, including the use of a momentum-updated history queue as in MoCo, selecting the topK nearest neighbours in optical flow in Eq.3, and computing a multi-instance InfoNCE loss. We will release all the source code later.

---

**Algorithm 1:** Pseudocode for CoCLR in PyTorch-like style.

```
# f_q, f_k: encoder networks for query and key, for RGB input
# g: frozen encoder network for Flow input
# f_q, g are initialized with InfoNCE weights
# queue_rgb: dictionary as a queue of K keys (CxK), for RGB feature
# queue_flow: dictionary as a queue of K keys (CxK), for Flow feature
# topk: number of Nearest-Neighbours in Flow space for CoCLR training
# m: momentum
# t: temperature
f_k.params = f_q.params # initialize
g.requires_grad = False # g is not updated by gradient

for rgb, flow in loader: # load a minibatch of data with N samples
        rgb_q, rgb_k = aug(rgb), aug(rgb) # two randomly augmented versions

        z1_q, z1_k = f_q.forward(rgb_q), f_k.forward(rgb_k) # queries and keys: NxC
        z1_k = z1_k.detach() # no gradient to keys

        z2 = g.forward(flow) # feature for Flow: NxC

        # compute logits for rgb
        l_current = torch.einsum('nc,nc->n', [z1_q, z1_k]).unsqueeze(-1)
        l_history = torch.einsum('nc,ck->nk', [z1_q, queue_rgb])
        logits = torch.cat([l_current, l_history], dim=1) # logits: Nx(1+K)
        logits /= t # apply temperature

        # compute similarity matrix for flow, Eq(3)
        flow_sim = torch.einsum('nc,ck->nk', [z2, queue_flow])
        _, topkidx = torch.topk(flow_sim, topk, dim=1)
        # convert topk indexes to one-hot format
        topk_onehot = torch.zeros_like(flow_sim)
        topk_onehot.scatter_(1, topkidx, 1)
        # positive mask (boolean) for CoCLR: Nx(1+K)
        pos_mask = torch.cat([torch.ones(N,1),
        topk_onehot], dim=1)

        # Multi-Instance NCE Loss, Eq(2)
        loss = - torch.log( (F.softmax(logits, dim=1) * mask).sum(1) )
        loss = loss.mean()

        # optimizer update: query network
        loss.backward()
        update(f_q.params)
        # momentum update: key network
        f_k.params = m*f_k.params+(1-m)*f_q.params

        # update dictionary for both RGB and Flow
        enqueue(queue_rgb, z1_k) # enqueue the current minibatch
        dequeue(queue_rgb) # dequeue the earliest minibatch
        enqueue(queue_flow, z2) # enqueue the current minibatch
        dequeue(queue_flow) # dequeue the earliest minibatch
```

---

# 3 Qualitative Results for Video Retrieval

Query            Top3 nearest neighbours

Figure 1: Nearest neighbour retrieval results with CoCLR representations. The left side is the query video from the UCF101 testing set, and the right side are the top 3 nearest neighbours from the UCF101 training set. CoCLR is trained only on UCF101. The action label for each video is shown in the upper right corner.

# References

[1] K. He, H. Fan, A. Wu, S. Xie, and R. Girshick. Momentum contrast for unsupervised visual representation learning. In *Proc. CVPR*, 2020.