[Reviews · NeurIPS 2020]

Review 1

Summary and Contributions: This paper proposed a novel method for video representation learning. Specifically, the method is based on the popular contrastive learning framework. By using the ground-truth class labels, the authors constructed an upperbound for NCE based methods and showed that "hard positives" are important but missing from these standard NCE based self-supervised learning methods. To tackle this, the authors set up two feature networks, one for RGB, another for optical flow, and use one network to search for "hard positives" while training the other one. The authors conducted experiments on tasks of action recognition and video retrieval to demonstrate their point. For action recognition, the authors carried out experiments on Kinetics-400, UCF101, HMDB and demonstrated promising results. For video retrieval, the authors conducted experiments on UCF and HMDB and show large performance improvements with the proposed method.

Strengths: + The upperbound experiment is convincing in showing the importance of capturing "intra-class" variation that's missing from recent methods based on instance discrimination (e.g. MoCo, SimCLR). + I like the idea of making use of video temporal information to learn "intra-class" variation in self-supervised learning. + I like the approach of harvesting hard positives through bootstrapping from the cycle of RGB and optical flow. + The authors have conducted experiments on 2 tasks (action recognition and video retrieval) with in total 3 datasets (Kinetics400, UCF101, HMDB). + The results, when compared to previous state-of-the-art methods, are competitive.

Weaknesses: - I think there is an important baseline that's missing from the evaluation. Since the proposed method utilizes both RGB and flow for self-supervised pretraining, the authors should train an InfoNCE model that takes in both RGB and optical flow as input (it could just be the concatenation of the two) and compare to this model. - L216-8, there is quite heavy augmentation during testing. Many other methods use only the center crop (e.g., XDC [2] and DPC [24]) or 3 spatial crops (left-center-right) and 10 uniform temporal crops. Could the author also try this evaluation setting out? - From the text, I can't find the details regarding to the training epochs for different methods in Table 1. I am curious are they all trained with the same number of epochs? (e.g., comparing InfoNCE Flow with 64.9 top-1 acc to CoCLR k=5 Flow with 68.7 top-1 acc) As it matters a lot for self-supervised learning to have a long training schedule (generally the longer the training, the better the performance) - L269, "and on K400 with K = 5 for 1 cycle only", is this because having 2 cycles will have deteriorate performance? If so, could you please provide some explanations/insights? - XDC[2] does provide pretraining results on Kinetics (74.2 for UCF and 39.0 on HMDB), it might be more informative to have this in Table 2 rather than XDC's IG65M results. - To provide a more comprehensive view on current state-of-the-art, I encourage the authors to add the following papers to Table 2, "Audio-Visual Scene Analysis with Self-Supervised Multisensory Features", "Audiovisual SlowFast Networks for Video Recognition", "Audio-visual instance discrimination with cross-modal agreement" and "Multi-modal Self-Supervision from Generalized Data Transformations".

Correctness: The approach/evaluation presented in this paper looks reasonable to me. I did not find any thing particularly wrong.

Clarity: The paper is well-written and easy to follow. I have a few minor suggestions though: - In L120-1, "confirming that instance discrimination is not making the best use of data". I can understand what the authors are trying to convey here, but I think this claim is too strong and the reasoning is flawed -- since UberNCE is using extra data (labels), to me it's not fair to compare InfoNCE and UberNCE and conclude InfoNCE is not making the best use of data. - In Section 3.2, is it possible to change all subscription "1" (for RGB) and "2" (for optical flow) to superscription "rgb" and "flow"? It would make the notation much more clear.

Relation to Prior Work: L70-7, there should have been some discussions about the relation to those work in the first two paragraphs of the related work section. (as per the policy "The related work section should not just list prior work, but explain how the proposed work differs from prior work appeared in the literature.")

Reproducibility: Yes

Additional Feedback: POST REBUTTAL I feel it's a paper with intuitive idea and good results from which people can learn new things. However, I can see some the concerns raised by other reviewers, and also I'm concerned about some of my comments not being properly addressed. Specifically, I want to point out 1) a baseline with RGB+flow as inputs is needed to demonstrate the effectiveness of the proposed method, and 2) it's inappropriate and misleading to call UberNCE as an "upper bound". It would make this paper more valuable if these can be taken into consideration in the revised version. That being said, overall I am still inclined towards accepting this paper. --------------------------------------------------------------- Please address my comments in the "weakness" section.


Review 2

Summary and Contributions: This paper presents a self-supervised approach to learn representations from video data. The authors propose to use the RGB and the flow "views" of the input video to train a representation using contrastive learning. The authors propose to use these two views of the data to find hard positive examples to improve the representation. The method is evaluated on action recognition datasets.

Strengths: + Self-supervised learning is a relevant topic for NeurIPS. This work focusses on video representations which in my opinion is one of the places where supervised learning doesn't shine, and indicates that we may need a different learning paradigm. Self-supervised learning is a possible alternative, and research in this direction should be encouraged. + Using complementary views of the data is useful for representation learning. In the case of videos, the authors show that using RGB and flow helps learn better representations. Although this result is known (Simonyan et al., 2014, Tian et al., 2019), I believe revisiting it in the case of self-supervised learning and demonstrating it conclusively on benchmark tasks is valuable. + Many recent methods focus on instances obtained via data augmentation as "positives" for contrastive learning. The authors show a way to use similarity ranking in the complementary view of the data to obtain positive sets that include more than a single instance. This idea seems intuitive and has not been shown for the case of video and optical flow. + The benchmark evaluation in this paper is sound. The authors compare against many prior methods (Table 2) on action recognition tasks.

Weaknesses: + The main claim of this paper is that co-training between RGB and flow learns better representations than just using RGB. There is prior well-cited work that demonstrates this fact as well (CMC - Tian et al., 2019). The authors do not compare against this work. It would not be a serious issue, but the following two reasons make this quite serious: - The learning objective proposed in CMC is (in my opinion) simpler than the current work and it is a few line of code changes from the authors' formulation. In the spirit of good empirical understanding, it is valuable to see what simple baselines achieve. - The authors propose a positive mining framework (similar to [A]) which is more involved than a simple multi-view contrastive objective. This paper does not answer the question whether the additional complexity is warranted. To sum up, the proposed method is intuitive, the results are good, but it is not clear if the co-training, positive mining, and alternate optimization steps (main contributions of this paper over prior work) are necessary. [A] - Audio-Visual Instance Discrimination with Cross-Modal Agreement

Correctness: There are missing evaluations in this paper that seriously question if the proposed improvements in this paper actually work. I have noted this down in the Weaknesses section.

Clarity: Yes. The paper is well written and is easy to understand. The authors explain their method and empirical evaluation setup well.

Relation to Prior Work: The paper glosses over some of the most related work (CMC) which also does RGB + Flow multi-view contrastive learning. While CMC is referenced, it is not discussed.

Reproducibility: Yes

Additional Feedback: The author response partially addresses my concern - is going beyond instance discrimination valuable, and how valuable is it? The comparison with CMC was critical to understand this hypothesis, and not having this in the paper made it hard to to accept. Do note that CMC has been around (open sourced, accepted at ICML) for a while, so not using it as a baseline makes it very hard to understand whether this paper and all the design decisions matter. In the author response, the authors only reported one setting (linear probing) and this isn't a very thorough analysis. Although incomplete, I am willing to change my rating to (6) since this is a step in the right direction. I agree with the analysis from R3 and R1 that the authors should discuss prior work better, and clarify their claims.


Review 3

Summary and Contributions: - The paper proposes an approach for self-supervised learning of representations from video data. - The paper first demonstrates that applying the NCE framework in a supervised learning setting leads to representations that are far superior to their self-supervised counterparts that are trained for instance classification. - The main approach proposed in this paper attempts to bridge the gap between the supervised and self-supervised representations by replacing the requirement of labelled data. - This is done by leveraging a `co-training' process: first, two domains of each sample (RGB and Flow here) are used to train separate representations in the NCE instance classification framework; then, the learned RGB representation is used to induce a clustering (pseudo-labeling) for the Flow representation and the learned Flow representation is used to induce a clustering for the RGB representation. This process is iteratively repeated to obtain the final representations. - The paper demonstrates results on downstream tasks of action recognition and video retrieval.

Strengths: - The paper is clearly written and organized well. The authors have made a good effort of presenting the contributions clearly. - The paper first demonstrates that applying the NCE framework in a supervised learning setting leads to representations that are far superior to their self-supervised counterparts that are trained for instance classification. The NCE supervised representations also outperform the representations that are trained with the more standard softmax cross-entropy objective. This result is novel (apart from the concurrent work [36]) and would provide an important insight to the NeurIPS community. - The idea of using a `co-training' approach to improve the NCE self-supervised learning framework is very interesting and novel. The motivation for using such an approach has been clearly motivated through UberNCE (supervised model) experiments. As also discussed in the paper, this idea is fairly general and can be applied to domains like video+text, video+audio, etc - The authors have chosen a good range of datasets for demonstrating the results on the downstream task of action recognition and video retrieval. However, I do have some concerns about the choice of tasks - discussed in the weaknesses. - The authors have conducted a large number of experiments on the chosen tasks and datasets. The results demonstrate that the supervised UberNCE model outperforms both the InfoNCE and Cross-entropy based models. The proposed co-training based approach CoCLR demonstrates significant gains over the InfoNCE baseline and comes relatively close to the performance of the UberNCE model. These inferences are made based on both the action recognition and the video retrieval tasks on the UCF101 dataset. - The authors also compare to numerous existing self-supervised learning approaches for performing the downstream tasks of action recognition. The results demonstrate that the proposed approach is comparable or better in most cases.

Weaknesses: # Related Work I believe that the related work section has been poorly executed. It simply lists numerous papers from 4 domains of existing research. This provides limited information about the position of the proposed work w.r.t these existing works. A more detailed discussion of the contrast between the proposed work and existing literature, or the similarities, or the parts that have been directly adopted is generally expected from the related work section. Furthermore, the authors may have missed the following highly relevant papers which are also discussed in my Baselines section below: 1) Yan, Xueting, et al. "ClusterFit: Improving Generalization of Visual Representations." Proceedings of the IEEE/CVF Conference on Computer Vision and Pattern Recognition. 2020. 2) Caron, Mathilde, et al. "Deep clustering for unsupervised learning of visual features." Proceedings of the European Conference on Computer Vision (ECCV). 2018. 3) Caron, Mathilde, et al. "Unsupervised pre-training of image features on non-curated data." Proceedings of the IEEE International Conference on Computer Vision. 2019. (If [1] was not published at the time of submission, please ignore) # UberNCE Upper Bound Claim The text mentions that "we establish an upperbound on the existing approaches trained with Noise Contrastive Estimation (NCE)" "we establish an upperbound for a linear probe based on contrastive learning by using UberNCE" and "Note that UberNCE is trained in a fully-supervised manner, and therefore acts as an upperbound for the contrastive learning of the visual representation under the linear probe protocol, for this dataset D" I do not believe that the performance of the UberNCE model is a true upper bound in any form. Self-supervised representations learned on a specific dataset do have the potential to outperform the fully-supervised models (like UberNCE or cross-entropy) trained on the same data. Therefore, the performance of these models is not really a bound (even in the linear probe setting). If this is not true, please provide an explanation for why that is the case. # Baselines While the authors have done a great effort of testing numerous variants of the proposed model, I believe that some important baselines may have been overlooked. 1) The CoCLR models are effectively trained for 800 epochs each, whereas the InfoNCE models have been trained for 300 epochs. We know from MOCOv2 and SimCLR that the performance of the learned representation seems to improve even beyond 300 epochs. I would suggest training baseline InfoNCE models upto 800 epochs each (RGB and Flow) to allow fair comparison (ensuring a similar learning rate schedule as advised in MOCOv2). 2) Is co-training really needed? Can just one of these domains be used in the same iterative fashion? For example, at 300 epochs, the fixed RGB representation can be used to form the positive sets for each sample and the representation can be further trained using these sets. This process can iteratively be done K times (same as CoCLR). 3) The experiment (2) suggested here is in similar vein as numerous existing self-supervised learning approaches that have recently been proposed [1],[2],[3] mentioned in the "Related Work" section above. The main idea is to simultaneously cluster the representations and use the clusters to learn improved representations. Therefore, the experiment in (2) would be important. # Additional experiments that would be good to add - The chosen tasks of action classification and information retrieval for presenting the main result in Table 1 are somewhat redundant. More specifically, a good action classification model (under a linear probe) clusters categories well and would result in a good retrieval performance (and vice versa). Ideally, a somewhat independent task should be chosen to demonstrate a complementary strength of the representation. Some suggestions would be - action detection (temporal localization), pose estimation (since actions are sequence of poses), video captioning. This is only a minor concern, especially since CoCLR seems to be outperforming numerous existing methods on the task of action classification across many datasets. - In the alternation stage, positive sets are chosen using the other representation. It would be great to see some visualizations of the sets chosen. This is especially important since the proposed method is supposed to act as a pseudo-labeling mechanism. If it indeed seems true from visualization that the sets are constructed according to action categories, this would be an interesting insight to add to the paper.

Correctness: 1) The claim of an upper bound on the performance of NCE models seems to be incorrect. I have mentioned this in the weaknesses section and I welcome discussion from the authors. 2) There seems to be a major difference in the training setting of the baseline InfoNCE model. The performance demonstrated by this model also seems extremely low "46.8" in Table 1 which might be attributed to the difference in number of training epochs.

Clarity: Yes, the paper is very well written and easy to follow. The paper has been organized well and contributions have been presented in a concise manner.

Relation to Prior Work: As mentioned in the weaknesses, in section "# Related Work", this discussion is severely limited in the text and some very relevant papers may have been overlooked.

Reproducibility: Yes

Additional Feedback: Post-Rebuttal: 1) Please make sure you do not claim UberNCE as an upper bound. Even though the supervised objective optimizes for optimal performance on a linear probe, not much can be said about it's generalization peformance. When you say upper bound, you are infact claiming that the generalization performance on a held-out set is optimal. This is NOT true! A self-supervised approach could have a representation that outperforms the supervised representation on the held-out set even in the setting of a linear probe. 2) Please add additional discussion to the related work section - as was pointed out in the initial review.


Review 4

Summary and Contributions: This submission proposes a co-training framework to learn representations from videos in a self-supervised way for video action recognition and video retrieval. The motivation is interesting, they want to mine hard positives from different views of data to enhance the representation power.

Strengths: 1. The motivation of paper is interesting. Instead of simply using optical flow for motion representation, they use optical flow as a way to mine hard positives when learning spatial stream based on RGB frames. 2. They introduce UberNCE to justify their claim that hard positives can benefit self-supervised learning. Experimental results support their claim well. 3. The experimental results on UCF101 and HMDB51 are reasonable.

Weaknesses: 1. The comparisons in Table 2 is not fair in terms of input. Most previous work do not use optical flow. We all know that optical flow can significantly improve a video action recognition model as demonstrated by the success of two-stream networks. I agree that optical flow still belongs to modality "V" (visual), but it is a very strong input representation. To emphasize, optical flow is more useful than audio and text modality on these video action recognition datasets. Given this situation, the performance of proposed CoCLR algorithm is not strong as it looks like in Table 2. 2. The framework is complicated and hard to reproduce. First of all, the proposed method requires computation of optical flow. Starting from year 2017, people shifted away from optical flow because its huge computational cost. Hence, the CoCLR algorithm is hard to scale. This makes it a big problem for self-supervised learning because the ultimate goal is to learn from unlimited video corpus. Second, as can be seen from line 201 to 209, the proposed method is a multi-stage pipeline, not end-to-end, and complicated (making it hard to reproduce). They need to first train RGB and Flow network for 300 epochs. Then do the alternation step for several cycles, each cycle contains 200 epochs. Compared to previous literature who only train the network on RGB frames for 100 or 200 epochs, the performance improvement might come from longer training. Recent publications such as SimCLR and MOCOv2 already shows that longer training help unsupervised representation learning a lot. 3. The performance is not strong as it looks like. I already partially mentioned it in point 1. In addition, I want to point out a concurrent work, Watching the World Go By: Representation Learning from Unlabeled Videos https://arxiv.org/abs/2003.07990 In this paper, they use 2D ResNet18 model, and can achieve 39% top-1 accuracy on Kinetics400 dataset, while this submission using R2+1D ResNet18 model only achieves 33.8% top-1 accuracy. At least to my knowledge, R2+1D is better than a 2D network for video action recognition. Hence, I have doubts on the performance of Kinetics400 dataset and the learned video representation. I understand this is an arxiv paper and you don't need to mention or compare to it. But could you provide some insights or discussions to clarify it? Thank you.

Correctness: Yes, they are correct to my best knowledge.

Clarity: Yes, it is well written.

Relation to Prior Work: No. Related work section is poorly done. Especially for self-supervised video representation learning, which is the main focus of this submission, the authors only use one sentence to describe them (line 72-73). The authors cite more than 20 papers in this field without discussing the difference between this work and previous work.

Reproducibility: No

Additional Feedback: Post-rebuttal: I appreciate the authors taking the time to write the rebuttal. I have read the reviews and authors' response. It partially addressed several of my concerns, but more needs to be done. I will keep my score as 4, which is on the negative side. To be specific, 1. Although authors promised to release their code and models, I still think the framework is too complicated and hard to reproduce and scale. For self-supervised representation learning, its goal is to learn effective representations from lots of unlabeled data and may need to update itself from time to time. However, the proposed CoCLR method is a multi-stage framework. It needs to mine the data first, then alternately train two networks and train each of them for multiple epochs. This creates the problem that there are many hyper-parameters to tune. Actually, as authors mentioned in the rebuttal, simply changing the lr schedule can improve the performance on Kinetics400 by 7 percent, which proves that this framework is sensitive to these hyper-parameters. I don't think this is a good sign for self-supervised representation learning. 2. Table 2 seems comprehensive, but as R3 and I pointed out, the related work section is poorly written. Without further clarification, it is hard to define the contributions from each component of the method, like co-training, positive mining and alternate optimization. 3. The proposed UberNCE is closely related to supervised contrastive learning [36], but no further discussion is presented in the paper. Besides, the claim of 'UberNCE' as an upper bound is inappropriate. And as pointed by R1, the comparison between infoNCE and uberNCE is not fair, thus the claim of the whole paper (instance discrimination is not making the best use of data) is too strong. Overall, the idea of the paper is interesting, but needs significant work to make it reasonable and good for publication. Hence, I will keep my score of 4. ---------------- 1. As I mentioned in weakness and reproducibility, CoCLR algorithm is a multi-stage pipeline and very complicated. I'm not confident that it can be reproduced smoothly. Besides, the authors mentioned a number of 33.8 accuracy for Kinetics400 in line 300, but without implementation details, like what is the learning rate, how many epochs, do they use both streams, etc. I would suggest to add more details on how authors perform the linear probe on Kinetics400. 2. Table 2 looks good, but actually not convincing. Adding another small table for fair comparison will make the submission stronger. For example, you can use DPC and SpeedNet as baseline, incorporating optical flow to their framework, and train for the same number of epochs as you do. In this way, readers can clearly see the benefits from CoCLR. It is important to dissect the improvement from each contribution you made. In the current stage of the submission, I'm not sure the performance improvement is coming from using optical flow and longer training, or from mining hard positives.

[Author Response · NeurIPS 2020]

We would like to thank all the reviewers for their constructive feedbacks, we will edit the paper to include the missing discussion and implementation details.

**[All Reviewers] Related work.**   We agree with the reviewers that a more extended discussion is required for related work, we will include all the suggested citations and discuss them properly in our final version.

**[R1,R3,R4] Additional baselines.   (1) Training infoNCE for more epochs.** As shown in Table 1, the effective training epochs for CoCLR-RGB and CoCLR-Flow were 500 epochs, ending up with 70.2% and 68.7% respectively for the linear probe. In contrast, training infoNCE for RGB and Flow for the same number of epochs (500), the linear probe results are only 52.3% (RGB) and 66.8% (Flow). **(2) Cross-domain co-training is effective.** Our CoCLR mines positives across RGB and optical flow domain. **R3** suggests comparing against the baseline that mines positives within the domain, similar to the deep clustering algorithm. We experiment with mining positives within the domain, specifically, using RGB representation from infoNCE to construct positive sets for each sample in the RGB domain. This model is trained on UCF101 with the same schedule as CoCLR for a fair comparison. The linear probe results on UCF101 is 48.7%, significantly lower than that of CoCLR-RGB (70.2%). This is due to the fact that such within-domain sample mining process will not provide *hard* positives, as the samples in the positive set are already close to its corresponding query sample in the same representation domain, so they are actually *easy* positives, violating our basic hypothesis on the important role of hard positives. However, with multiple domains, easy positives in one domain are very likely to be hard in the other domain. **(3) Compare against other methods with optical flow input.** We note a recent arXiv paper (MemDPC, to appear in ECCV2020 by Han *et al.*) has also used both RGB and optical flow, yet CoCLR still outperforms it on both classification (90.7% vs. 86.1%, for pretrain on K400 and finetune on UCF101) and retrieval tasks (55.9% vs. 40.2% on R@1 on UCF101). We will add these discussions.

**[R1,R3] UberNCE upperbound claim.**   UberNCE is in fact a supervised learning objective that encourage samples from the same category to be well-clustered, and samples from different categories to be separated on a unit sphere (forming spherical caps). Intuitively, this will enable the classes to be *linearly separable* in the feature space, which is the common criterion for evaluating representation quality. We will rephrase this claim.

**[R1] (1) Test time augmentation.** We actually used the same augmentation as DPC in their released codebase. **(2) Training cycles.** For the K400 experiments, we only managed to finish 1-cycle of training at the time of submission, but we conjecture more cycles will be beneficial, we will add the n-cycle results in the final version.

**[R2] Relation and comparison to CMC.   (1) Difference.** CMC is an excellent paper, but its training scheme is fundamentally different to that of CoCLR. CMC aims to maximize the mutual information between **multiple domains of the same video clip**, *i.e.* learning the correspondence between its RGB and Flow representations, thus, the proxy task defined by CMC is still limited to be *instance discrimination*. However, CoCLR goes beyond this by explicitly allowing **multiple instances from the same domain** to be positive pairs, specifically, the most similar clip to one query sample based on flow representation is treated as a positive pair in the RGB domain, this design allows to exploit the complementary nature between different modalities, in addition, we also adopts a noise-tolerant MIL-NCE loss. These are the core contributions of our paper. **(2) Experiments.** We note that CMC will appear at ECCV2020, so it is *not* officially published yet. Nevertheless, we have carried out a comparison on UCF101, by training a CMC model with the same backbone (S3D) as ours for the same number of epochs (fair comparison with CoCLR). Specifically, two networks for both RGB and Flow streams are trained simultaneously by maximizing the mutual information among three views $\text{RGB}_t, \text{RGB}_{t+k}, \text{Flow}_t$, as in CMC section 4.2. We evaluated the RGB model by **linear probing**: the CMC model gets 55.2% on UCF101 (note that, the model in original CMC paper actually finetunes the entire network, and only get 59.1%, this is significantly lower than all of our finetune models in Table 1, above 78%), whereas the CoCLR-RGB model gets 70.2% by linear probing. We hope this has resolved the major concern raised by R2.

**[R4] (1) CoCLR performance relies on flow as input.** As shown in Table2, our CoCLR-RGB achieves 87.3% top1 accuracy, even without using flow for inference. Considering the size of the training data, we argue that CoCLR is amongst the most efficient and effective approaches. In addition, the general approach of co-training idea is not limited to flow (L64-68), and it remains unclear to us why flow is a stronger representation than text – after all, one can treat supervised learning as one special case of data with a text modality. **(2) Reproducibility.** This concern can be **fully resolved** by releasing all the training codes and models, we have promised to do so in the draft (as mentioned in L328). **(3) Training epochs.** CoCLR is trained longer only on the small dataset (UCF101) for the ablation study. But for K400, CoCLR was trained for 150 epochs (100 for infoNCE and 50 for the 1st cycle), we will clarify this. Furthermore, measuring epochs is only meaningful when all approaches use the same dataset for training. As shown in Table 2, other competitive approaches have been trained on orders of magnitude more data than K400, *e.g.* IG65M ($273\times$), HTM ($196\times$), Youtube8M-2 ($169\times$), 1 epoch on these datasets will be equivalent to over 100 epochs on K400. **(4) K400 linear probe result.** The evaluation from "Watching the World Go By" mentioned by R4 actually trains both **a LSTM and a linear layer** to get 39% accuracy, so it is *not* a linear probe. At the time of submission, we reported 33.8% linear probing accuracy on K400 with RGB input. We have re-evaluated the same CoCLR-RGB representation with a proper regularization and learning rate schedule, achieving **40.5** top1-accuracy. In addition, we will continue CoCLR training on K400 for more epochs (match other competitive approaches) and this accuracy is expected to be further boosted.

[Meta-Review · NeurIPS 2020]

This paper presents an approach to learn video representation via contrastive learning framework. All the reviewers like the proposed approach calling it intuitive and a step in right direction. Several concerns were there as well: (a) comparison to CMC; (b) relation to prior work; (c) reproducibility; (d) UberNCE being upper-bound. Authors submitted a strong rebuttal.They provided comparisons to CMC, promised to do better discussion of related work, release code. UberNCE argument still remains a concern but the this is a simple change. AC agrees with the reviewers and recommends acceptance. Please make all the changes suggested by reviewers in camera ready.